# PREFERENCE LEARNING UNLOCKS LLMS' PSYCHO-COUNSELING SKILLS

## ABSTRACT

Applying large language models (LLMs) to assist in psycho-counseling is an emerging and meaningful approach, driven by the significant gap between patient needs and the availability of mental health support. However, current LLMs struggle to consistently provide effective responses to client speeches, largely due to the lack of supervision from high-quality real psycho-counseling data, whose content is typically inaccessible due to client privacy concerns. Furthermore, the quality of therapists' responses in available sessions can vary significantly based on their professional training and experience. Assessing the quality of therapists' responses remains an open challenge. In this work, we address these challenges by first proposing a set of professional and comprehensive principles to evaluate therapists' responses to client speeches. Using these principles, we create a preference dataset, PsychoCounsel-Preference, which contains 36k high-quality preference comparison pairs. This dataset aligns with the preferences of professional psychotherapists, providing a robust foundation for evaluating and improving LLMs in psycho-counseling. Experiments on reward modeling and preference learning demonstrate that PsychoCounsel-Preference is an excellent resource for LLMs to acquire essential skills for responding to clients in a counseling session. Our best-aligned model, PsychoCounsel-Llama3-8B, achieves an impressive win rate of 87% against GPT-4o. We will release PsychoCounsel-Preference, PsychoCounsel-Llama3-8B and the reward model PsychoCounsel-Llama3-8B-Reward to facilitate the research of psycho-counseling with LLMs.

## 1 INTRODUCTION

According to the World Mental Health Report[1], the global demand for mental health support far exceeds the availability of accessible, affordable, and timely care. Millions of individuals struggle with mental health challenges, yet many face a severe shortage of trained professionals, particularly in low- and middle-income countries. The emergence of Large Language Models (LLMs) has accelerated the integration of AI into psycho-counseling, thanks to their remarkable ability to comprehend human intent and provide effective responses (Ouyang et al., 2022). However, due to the complexity of clients' situations and the professional skills required, current LLMs still face challenges in consistently providing effective responses to client speeches during counseling sessions (Na et al., 2025; Zhang et al., 2024; Chung et al., 2023).

The sensitive and private nature of counseling sessions poses significant challenges for obtaining publicly available datasets that accurately reflect real-world interactions (Stade et al., 2024). This scarcity of resources hinders efforts to train LLMs in effectively understanding and responding to client speech within counseling contexts. Moreover, the quality of responses provided by therapists can vary widely, influenced by their level of professional training and experience (Rocco et al., 2019). This variability impacts the consistency of counseling effects and underscores the importance of standardizing and assessing therapist responses.

To address these gaps, we collaborated with professors in social work and psychiatry to develop a set of professional and comprehensive principles for evaluating therapists' responses to client speeches. These principles assess not only the fundamental aspects of a response in a counseling session, such

---

[1]https://www.who.int/teams/mental-health-and-substance-use/world-mental-health-report

as empathy, relevance, conciseness, and safety, but also extend the effectiveness of a response based on professional psycho-counseling theory. This includes evaluating whether the response promotes clients' self-exploration, enhances their autonomy, and effectively identifies the client's stage of change.

Using these principles, we extract high-quality responses from the generations of a pool of popular LLMs and construct a large psycho-counseling preference dataset, PsychoCounsel-Preference. The dataset comprises 26,483 unique client speeches spanning 8 coarse-grained and 42 fine-grained topics. To the best of our knowledge, PsychoCounsel-Preference is the first large-scale and comprehensive psycho-counseling preference dataset. We hired professional psychotherapists for verification, and their annotations exhibit strong agreement within PsychoCounsel-Preference, ensuring the dataset's reliability and consistency.

Experiments show that our reward models trained with PsychoCounsel-Preference show an excellent ability of evaluating responses to clients while previous start-of-the-art reward models lag behind. Moreover, we apply both online and offline preference learning on PsychoCounsel-Preference or the trained reward models. Our best resulting model, PsychoCounsel-Llama3-8B, achieves the state-of-the-art performance on the testing set of PsychoCounsel-Preference, with an impressive **win rate of 87% against GPT-4o**. Feedback from professional psychologists shows that PsychoCounsel-Llama3-8B could give more balanced and desirable responses under length constraint during the inference stage. Through further analysis and case study, we demonstrate the advantage of training online over offline and provide insights into how to improve the model performance in the future.

## 2 RELATED WORK

### 2.1 LLMs ASSISTING PSYCHOTHERAPY

Integrating LLMs into Psychotherapy is not a trivial process which could articulated as a continuum of stages of assistive AI, collaborated AI, and fully autonomous AI (Stade et al., 2024). Currently, we are still in the first two stages where models operating tasks need human supervision. Related tasks include cognitive disorder detection (Shreevastava & Foltz, 2021; Chen et al., 2023b), negative thoughts recognition and reframing (Maddela et al., 2023; Sharma et al., 2024), and patient simulation (Chen et al., 2023a) or therapist simulation Liu et al. (2023), among which therapist simulation is the primary goal across the stages. However, Zhang et al. (2024) found that due to the lack of public high-quality data in psychotherapy and the complexity of clients' situation, LLMs still are not able to give effective responses to a client's speech consistently in a therapy session. Our work focuses on psycho-counseling, which is a short-term, supportive process for helping individuals cope with life challenges and emotional distress, and constructs a high-quality preference dataset that aligns with the preference of professional psychotherapists and uses preference learning to train helpful and reliable assistants for psycho-counseling.

### 2.2 HUMAN PREFERENCE ALIGNMENT

Human preference alignment has been shown to be a critical step in making LLMs helpful, harmless, and honest (Ouyang et al., 2022; Bai et al., 2022a). Current methods can be broadly categorized into offline and online approaches. Offline methods optimize the model using a pre-annotated set of preference data with objectives such as DPO (Rafailov et al., 2023). Online methods, on the other hand, generate outputs during training and utilize a reward function to score them. High-scoring generations are encouraged, while low-scoring ones are discouraged through policy gradient methods such as PPO (Schulman et al., 2017).

Compared to offline alignment, online methods are more computationally expensive and require careful hyperparameter tuning to ensure stable training (Xu et al., 2024). Offline methods, which frame alignment as optimizing a classification loss, eliminate the need for a reward model, making them more stable and efficient. However, they are susceptible to distribution shifts (Marks et al., 2023). Furthermore, Tang et al. (2024) found that optimizing with online preferences instead of offline data can lead to better model performance. Iterative direct preference learning combines the strengths of both offline and online methods. In this approach, preference data is generated online and used to optimize an offline learning objective (Pang et al., 2024), which has been demonstrated as a strong baseline in both academia (Xu et al., 2024) and industry (Yang et al., 2024).

# 3    PSYCHOCOUNSEL-PREFERENCE

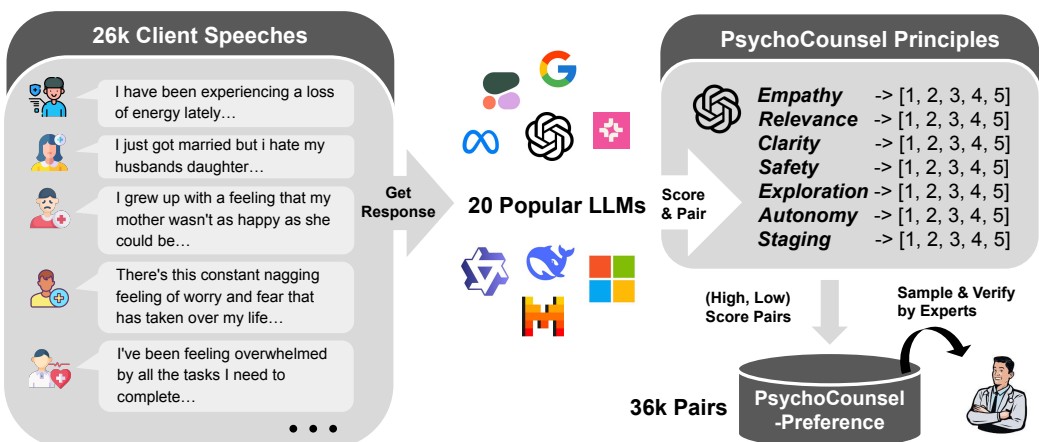

Figure 1: PsychoCounsel-Preference Construction Pipeline. 1) We first collect over 26k client speeches covering a wide range of topics from various sources, applying necessary data cleaning. 2) 20 popular LLMs are sampled and prompted to roleplay as psychotherapists and give responses to these client speeches. 3) GPT-4o is instructed to evaluate the responses based on our proposed PsychoCounsel Principles, and preference pairs with substantial score gaps are incorporated into PsychoCounsel-Preference.

## 3.1    CLIENT SPEECH COLLECTION

We collect client speeches from various data sources: counsel-chat[2], MentalAgora (Lee et al., 2024), TherapistQA (Shreevastava & Foltz, 2021), Psycho8k (Liu et al., 2023), and several huggingface datasets (amod-counsel[3], MentalChat16K[4], and phi2Mental[5]). Client speeches with number of characters more than 1,000 and less than 100 are discarded to ensure a proper length of context. After an additional step of de-duplication, the resulting data contains 26,483 client speeches with average length of 366 characters covering a wide range of topics including 8 coarse topics: Core Mental Health Issues (9,054), Emotional Well-being and Coping Strategies (5,717), Relationships and Interpersonal Dynamics (6,483), Life Transitions and Challenges (934), Social Issues (667), Youth and Development (1,175), Crisis and Safety Concerns (529) and Special Topics (1,924). Under these 8 topics are 42 fine-grained topics (see Table 5 in the appendix for the detailed topic distribution).

## 3.2    PSYCHOCOUNSEL PRINCIPLES

To answer the question *what is a good response to a client speech in psycho-counseling*, we collaborate with experts in social work and psychiatry (our co-authors) and propose a set of professional principles to measure the response to a client speech from seven different dimensions:

---

[2]https://github.com/nbertagnolli/counsel-chat

[3]https://huggingface.co/datasets/Amod/mental_health_counseling_conversations

[4]https://huggingface.co/datasets/ShenLab/MentalChat16K

[5]https://huggingface.co/datasets/saxenaindresh681/microsoft-phi2-mental-health

> **PsychoCounsel Principles**
>
> **Empathy and Emotional Understanding**: The response should convey genuine empathy, acknowledging and validating the client's feelings and experiences.
> **Personalization and Relevance**: The response should be tailored to the client's unique situation, ensuring that the content is directly relevant to their concerns.
> **Clarity and Conciseness**: The response should be clear, well-organized, and free of unnecessary jargon, making it easy for the client to understand and engage with.
> **Avoidance of Harmful Language or Content**: The response should avoid any language or content that could potentially harm, distress, or trigger the client, ensuring the interaction is safe and supportive.
> **Facilitation of Self-Exploration**: The response should encourage the client to reflect on their thoughts and feelings, promoting self-awareness and insight.
> **Promotion of Autonomy and Confidence**: The response should support the client's sense of control over their decisions and encourage confidence in their ability to make positive changes.
> **Sensitivity to the Stage of Change**: The response should recognize the client's current stage in the process of change and address their needs accordingly. If the client is in an early stage—uncertain or ambivalent about making a change—the response should help them explore their thoughts and motivations. If the client is in a later stage and has already made changes, the response should focus on reinforcing progress, preventing setbacks, and sustaining positive outcomes.

Please refer to Box 8 for the complete definition of the principles. Among these seven principles, **Facilitation of Self-Exploration**, **Promotion of Autonomy and Confidence**, and **Identifying Stages and Reasons for Change** emphasize a client-centered approach, which is recognized as a hallmark of effective psycho-counseling (Miller & Stephen Rollnick; Rooney et al., 2017; Hogarty, 2002; Tower, 1994). We use these three principles to measure the *effectiveness* of a response to a client speech, complementary to the other four principles, which are more basic requirements for an AI, requiring the response to be *empathy*, *relevant*, *concise*, and *safe* (Bai et al., 2022b; Ouyang et al., 2022). Evaluating therapist responses using these fine-grained principles provides a more structured and nuanced assessment of their effectiveness. Unlike general evaluations that focus solely on overall quality, this detailed approach allows for a deeper understanding of how well a response supports the client's emotional and psychological needs.

### 3.3 PREFERENCE GENERATION

We apply the generate-score-pair pipeline as Cui et al. (2023) to construct the PsychoCounsel-Preference dataset. For each client speech, we randomly sample four off-the-shelf LLMs from a model pool to give the response and instruct GPT-4o to annotate each response with 5-Likert scores for each principle defined in Section 3.2; higher scores mean more alignment with the principles. Then scores of the principles are averaged to get the overall score for a response and preference pairs are generated based on the overall scores. The whole pipeline is illustrated in Figure 1. To increase the diversity of the model responses, we initialize the model pool with 20 popular LLMs of a range of sizes developed by different organizations shown in Table 6. We also include LLMs with different architectures other than pure transformers like AI21-Jamba-1.5-Mini (Jamba Team et al., 2024), which is a hybrid transformer-mamba model. We randomly held out 3,291 client speeches for testing and the remaining 23,192 for training. After obtaining the scores of principles, for training, we extract response pairs with the overall score gap larger than or equal to 1 as the preference pairs, and for testing, we only extract the ([highest score response], [lowest score response]) pairs and pairs with the score gap less than 1 are discarded. In this way, we could exclude response pairs with similar scores, whose quality may be hard to differentiate.

Ultimately, PsychoCounsel-Preference includes 34,329 training preference pairs and 2,324 testing pairs. The models most likely to be chosen and those most likely to be rejected vary significantly in size (see Figures 5 and 6 for the distributions of chosen and rejected models). This suggests that simply scaling model size is not a decisive factor in making LLMs effective responders in psycho-counseling. We also observe that LLMs developed by non-English-speaking institutions are more likely to be rejected compared to those from English-speaking countries. This may suggest that non-English-speaking institutions have a greater need to enhance the capabilities of LLMs in their

respective languages, potentially leading to less emphasis on developing psycho-counseling skills in English.

### 3.4 PREFERENCE VALIDATION

To validate the quality of synthetic human preferences in PsychoCounsel-Preference, we hired two professional psychotherapists through Upwork[6] and instructed them to annotate preferences based on each principle and give the overall preference. The annotation set consists of 200 preference pairs randomly sampled from PsychoCounsel-Preference. The two therapists agree on 174 out of 200 samples. Additionally, one expert's annotations align with the preference labels in PsychoCounsel-Preference for 184 out of 200 samples, while the other aligns for 170 out of 200 samples. These results indicate a high level of agreement between the experts (87%) and demonstrate strong alignment between the expert annotations and the preference labels in PsychoCounsel-Preference (88.5%). This strongly suggests that the labels in PsychoCounsel-Preference are reliable and trustworthy.

## 4 EXPERIMENTS

### 4.1 REWARD MODEL

| Model | Acc. | AUC ($\uparrow$) | ECE ($\downarrow$) | Brier ($\downarrow$) |
|---|---|---|---|---|
| **State-of-the-art Reward Models** | | | | |
| Skywork-Reward-Llama-3.1-8B-v0.2 (Liu et al., 2024) | 57.9 | 0.623 | 0.331 | 0.379 |
| Skywork-Reward-Gemma-2-27B (Liu et al., 2024) | 69.2 | 0.740 | 0.123 | 0.229 |
| Llama-3.1-Nemotron-70B-Reward (Wang et al., 2024) | 87.3 | 0.938 | **0.040** | 0.102 |
| **Generative LLMs** | | | | |
| gemma-2-9b-it (Gemma Team, 2024) | 81.5 | - | - | - |
| Mistral-Nemo-Instruct-2407[7] | 78.0 | - | - | - |
| Llama-3.1-8B-Instruct | 80.1 | - | - | - |
| Llama-3.1-70B-Instruct (Llama Team, 2024) | 88.2 | - | - | - |
| **Our Reward Models** | | | | |
| PsychoCounsel-Llama3-3B-Reward | **98.1** | 0.997 | 0.050 | **0.014** |
| PsychoCounsel-Llama3-8B-Reward | 97.8 | **0.998** | 0.045 | 0.016 |

Table 1: Performance on the Testing Set of PsychoCounsel-Preference

Following Ouyang et al. (2022) and Bai et al. (2022a), we train Bradley-Terry (BT) style reward models $r_\theta(\cdot)$ where a linear head added on the top of LLMs outputs a scalar reward. Given a pair of preference data $\{y_c, y_r\}$ to a prompt $x$, the objective is to optimize the reward gap between chosen response $y_c$ and rejected response $y_r$:

$$\mathcal{L} = -\log\left(\sigma\left(r_\theta(x, y_c) - r_\theta(x, y_r)\right)\right),$$

where the sigmoid function $\sigma(\cdot)$ generates the probability of $y_c$ preferred than $y_r$.

We use Llama3.2-3B-Instruct and Llama3.1-8B-Instruct (abbreviated as Llama3-3B and Llama3-8B) to initialize the BT models, training them on PsychoCounsel-Preference for 2 epochs with a batch size of 128 and a learning rate of 9e-6. To evaluate our reward models, we compare them against three state-of-the-art reward models that rank highly on RewardBench (Lambert et al., 2024), as well as three popular LLMs, which are prompted to rank responses (see Box 7 for the prompt).

The overall results on PsychoCounsel-Preference testing response pairs are shown in Table 1. Our reward models significantly outperform all other reward models and generative LLMs, achieving notably high accuracy and ROC AUC Score (Bradley, 1997) on the PsychoCounsel-Preference testing set. These results suggest that PsychoCounsel-Preference provides robust supervision for training powerful reward models capable of effectively ranking responses to client speeches. We also

---

[6]https://www.upwork.com/
[7]https://mistral.ai/news/mistral-nemo/

| Model | Training Data | Acc. | AUC (↑) | Brier (↓) | ECE (↓) | Acc. (RewardBench) |
|---|---|---|---|---|---|---|
| Llama-3B | HelpSteer2 | 81.6 | 0.916 | 0.120 | **0.044** | 83.6 |
| | HelpSteer2 + Ours | **97.6** | **0.998** | **0.017** | 0.045 | **86.1** |
| Llama-8B | HelpSteer2 | 81.7 | 0.898 | 0.128 | **0.019** | 86.6 |
| | HelpSteer2 + Ours | **97.5** | **0.998** | **0.018** | 0.040 | **87.2** |

Table 2: Ablation study on reward models trained with HelpSteer2 vs. HelpSteer2 + our PsychoCounsel-Preference dataset. Results are reported on our test set and RewardBench.

calculate the Expected Calibration Error (ECE) (Naeini et al., 2015) and Brier Score[8] to assess the calibration level of the models. The results demonstrate that our reward models have comparable and low ECE values to the state-of-the-art reward model, Llama-3.1-Nemotron-70B-Reward, while achieving significantly better Brier Scores. This indicates that our reward models could give more reliable rewards and preference probability.

To further validate the effectiveness of our reward models, we train Llama3-3B and Llama3-8B under two settings: (i) on HelpSteer2 (Wang et al., 2024), a general-domain preference dataset, and (ii) on a merged dataset consisting of HelpSteer2 and our newly developed PsychoCounsel-Preference. For evaluation, we use both our PsychoCounsel-Preference test set and RewardBench (Lambert et al., 2024), a widely adopted general benchmark for reward models. The results, presented in Table 2, demonstrate that incorporating PsychoCounsel-Preference consistently improves performance across both Llama model sizes and the gains are not confined to domain-specific test set but also generalize to RewardBench. This indicates that PsychoCounsel-Preference provides complementary supervision to HelpSteer2 and enhances reward modeling in both in-domain and out-of-domain settings.

## 4.2 POLICY MODEL

| Model | Mental Issues | Emotional Well-being | Relation-ships | Life Changes | Social Issues | Youth | Safety | Special Topics | Overall |
|---|---|---|---|---|---|---|---|---|---|
| | | | | w/o Length Constraint | | | | | |
| Llama3-3B | 30.5 | 27.1 | 26.7 | 30.4 | 28.4 | 24.7 | 36.9 | 28.9 | 28.5 |
| + DPO | 57.7 | 59.1 | 57.9 | 63.5 | 54.3 | 53.4 | 60.0 | 64.7 | 58.5 |
| + DPO-Iter | 66.7 | 70.9 | 68.7 | 75.7 | 70.4 | 65.8 | 75.4 | 77.9 | 69.4 |
| Llama3-8B | 28.9 | 31.5 | 28.0 | 33.9 | 23.5 | 29.5 | 33.8 | 26.8 | 29.3 |
| + DPO | 70.2 | 74.5 | 73.7 | 74.8 | 77.8 | 73.3 | 80.0 | 74.0 | 72.9 |
| + DPO-Iter | **86.3** | **88.2** | **87.1** | **87.0** | **91.4** | **87.0** | **90.8** | **84.3** | **87.0** |
| | | | | w/ Length Constraint | | | | | |
| Llama3-3B | 15.1 | 15.5 | 15.0 | 15.7 | 13.6 | 13.7 | 9.20 | 15.3 | 15.0 |
| + DPO | 36.8 | 38.1 | 35.6 | 37.4 | 42.0 | 39.7 | 30.8 | 37.9 | 37.0 |
| + DPO-Iter | 47.5 | 46.7 | 46.3 | 40.9 | 46.9 | 45.2 | 47.7 | 43.4 | 46.4 |
| Llama3-8B | 19.3 | 17.9 | 17.1 | 21.7 | 21.0 | 17.1 | 18.5 | 19.6 | 18.5 |
| + DPO | 50.4 | 48.0 | 47.3 | 53.9 | 45.7 | 54.1 | 46.2 | 51.9 | 49.3 |
| + DPO-Iter | **75.6** | **77.9** | **79.2** | **77.4** | **76.5** | **73.3** | **83.1** | **74.0** | **77.0** |

Table 3: Win rates (%) of models trained with two different methods (+DPO or +DPO-Iter) on PsychoCounsel-Preference under two different settings (w/o and w/ Length Constraint) compared to GPT-4o. **Bold** numbers indicate the best performance in each column for each setting. The overall win rates of models that outperformed GPT-4o are underlined.

To further verify the effectiveness of PsychoCounsel-Preference and the trained reward models, we employ two preference alignment methods to optimize base models. **1) DPO**: we directly optimize the DPO (Rafailov et al., 2023) objective on PsychoCounsel-Preference:

---

[8]https://en.wikipedia.org/wiki/Brier_score

$$\mathcal{L}_{\text{DPO}}(\pi_\theta; \pi_{\text{ref}}) = -\mathbb{E}_{(x, y_c, y_r) \sim \mathcal{D}} \left[ \log \sigma \left( \beta \log \frac{\pi_\theta(y_c \mid x)}{\pi_{\text{ref}}(y_c \mid x)} - \beta \log \frac{\pi_\theta(y_r \mid x)}{\pi_{\text{ref}}(y_r \mid x)} \right) \right].$$

**2) DPO-Iter**: we follow an iterative approach (Pang et al., 2024), where, in each iteration, 8 responses are generated for each client speech and ranked by the reward model of the same size as the base model. The responses with the highest and lowest rewards are then annotated as online preference pairs, which are used to train the base model with the DPO objective. The client speeches for each iteration are 6400 sampled from the train set of PsychoCounsel-Preference. We use Llama3.2-3B-Instruct and Llama3.1-8B-Instruct as the base models. The training configuration includes a batch size of 64, a learning rate of 5e-7, and a total of 1,600 training steps. A development set comprising 10% of the training set from PsychoCounsel-Preference is used to select the best checkpoints. We set the value of $\beta$ as 0.1 for DPO across all the experiments.

**Evaluation** We use LLM-as-judge (Zheng et al., 2023) to effectively approximate human preferences (validated by the human experts) for evaluation. We prompt the model to generate responses for the testing client speeches in PsychoCounsel-Preference and leverage GPT-4o to compare these responses against those of GPT-4o using the proposed PsychoCounsel Principles. Specifically, we prompt the model in two settings: **1) w/o Length Constraint**: The models are instructed to act as therapists and respond to the given client speech without any restrictions on response length. **2) w/ Length Constraint**: To ensure a fairer comparison with GPT-4o, we impose a length constraint, requiring the models to generate responses of similar length to those produced by GPT-4o. The overall win rates of the models against GPT-4o are calculated for comparison. We also show the win rates for the coarse topic categories.

**Main Results** As shown in Table 3, in the w/o Length Constraint setting, the base models have low probabilities of outperforming GPT-4o. However, the models after alignment demonstrate significantly higher win rates against GPT-4o, indicating that supervision from PsychoCounsel-Preference effectively guides the models in learning how to respond to client speeches. Notably, Llama3-8B(+DPO-Iter) achieves the best performance, with a high overall win rate of **87.0% against GPT-4o**. This result suggests that online training and larger model sizes can potentially enhance generation quality, and **models with approximately 8B parameters can effectively develop the skills to respond to client speeches under the guidance of reward models trained on PsychoCounsel-Preference**. Compared to models in the w/o Length Constraint setting, those in the w/ Length Constraint setting generally have lower win rates against GPT-4o. We attribute this to the stricter generation constraint, which requires our models to align their response length with that of GPT-4o. However, our model, Llama3-8B (+DPO-Iter), still achieves a high win rate of 77% against GPT-4o, demonstrating that with proper training, the model can develop a robust ability to effectively respond to clients, regardless of generation constraints such as response length. We refer to the best model Llama3-8B (+DPO-Iter) as PsychoCounsel-Llama3-8B.

**Human Evaluation** We instruct the hired psychotherapists to provide preference judgments between the 200 randomly sampled response pairs generated by PsychoCounsel-Llama3-8B and GPT-4o, among which 100 for w/o Length Constraint setting and 100 for w/ Length Constraint. The provided order is shuffled to eliminate any position bias in the evaluation. In 82.5% of cases, GPT-4o and human experts made the same judgments, indicating that **GPT-4o serves as a reliable evaluator for assessing psycho-counseling responses**. Figure 2 presents the human experts' comparison between the two models based on the annotation principles. Overall, **real experts clearly prefer the outputs of PsychoCounsel-Llama3-8B across both evaluation settings and nearly all principles**. Only if no length constraint is applied, PsychoCounsel-Llama3-8B exhibits lower clarity compared to GPT-4o. This is primarily because PsychoCounsel-Llama3-8B tends to generate longer responses, which aligns with the observed phenomenon that as LLMs develop more complex capabilities through reinforcement learning (RL), they tend to produce more tokens (DeepSeek-AI et al., 2025). However, in the w/ Length Constraint setting, where models generate responses of similar length, PsychoCounsel-Llama3-8B demonstrates better performance in *Clarity*, *Safety* and *Staging*. This suggests that applying a length constraint after RL training is a promising approach to obtain more balanced and desirable generations.

Additionally, higher win rates only indicate the **relative** quality of responses. To provide an **absolute** assessment of the responses generated by PsychoCounsel-Llama3-8B, we instructed experts to assign fine-grained scores based on the PsychoCounsel Principles. Figure 3 presents the average scores

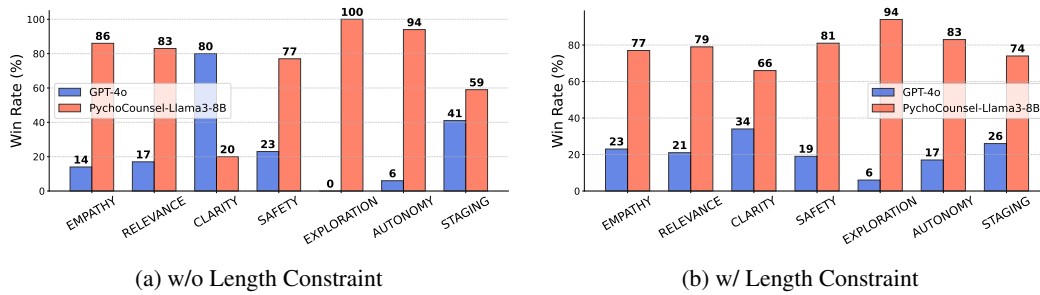

(a) w/o Length Constraint

(b) w/ Length Constraint

Figure 2: Experts' Comparison between GPT-4o and PychoChat-Llama3-8B in Two Settings

of 100 randomly sampled responses, evaluated by experts under two different settings. We observe that, except for *Clarity* in the w/ Length Constraint setting, PsychoCounsel-Llama3-8B achieves consistently high scores ($>4$) across all principles, indicating a strong alignment with the criteria for effective responses in psycho-counseling.

## 4.3 ABLATION STUDY

To explore the differences between training on offline vs online data and base models with different sizes, we set up a controlled experimental group. In this setup, base models are trained by DPO with two different sets of preference data, one is offline preferences from PsychoCounsel-Preference, and the other is trained using online preferences generated by the base model. Responses are selected by the reward model of the same size trained on PsychoCounsel-Preference. All the other experimental settings are kept identical with the training epoch as 1, learning rate as 5e-7, and global batch size as 64. Figure 4 illustrates the win rates of checkpoints against GPT-4o on the testing client speeches of PsychoCounsel-Preference.

In general, training on online samples demonstrates clear advantages over offline training: **1) Better Performance**: For Llama3-3B, training with online data (green line) consistently achieves a higher win rate compared to training with offline data (orange line). Similarly, for Llama3-8B, training with online data (blue line) delivers performance comparable to the best checkpoints of training with offline data (red line). **2) Greater Stability**: Both offline training curves exhibit a pronounced hump-shaped pattern, a known indicator of reward hacking (Rafailov et al., 2024). In contrast, the performance of online training models remains more stable across training steps. **3) Enhanced Compatibility with Smaller LLMs**: Online samples enable Llama3-3B to perform on par with, and sometimes even surpass, Llama3-8B models. This highlights the significant potential of combining online training with smaller LLMs. We can also observe that in this setting, training online can be viewed as DPO-Iter with only one round of updates on the policy model. However, its performance lags significantly behind DPO-Iter (shown in Table 3), indicating that utilizing online generations from the latest updated policy is crucial for more effective online preference learning.

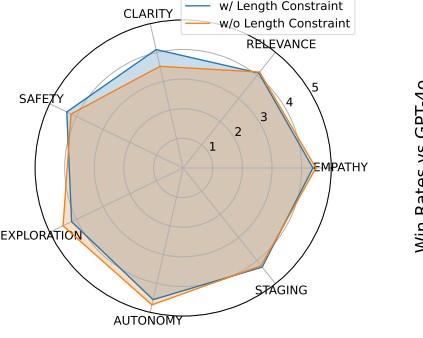

Figure 3: Absolute Scores

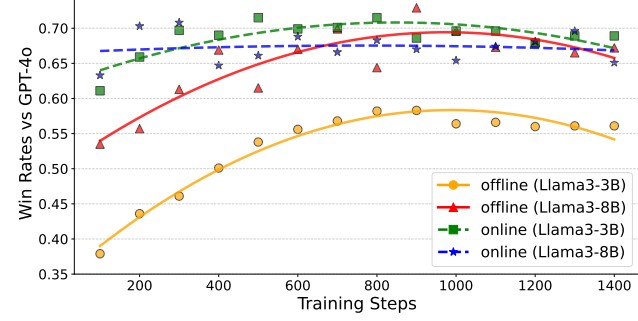

Figure 4: Comparison of Training Online or Offline

## 4.4 CASE STUDY

We sample testing cases and compare the outputs of GPT-4o with those of our best policy model, PsychoCounsel-Llama3-8B. Generally, PsychoCounsel-Llama3-8B provides better responses than GPT-4o. Table 4 provides a typical example in the w/ Length Constraint setting. We could see that the response of PsychoCounsel-Llama3-8B, not only validates the client's distress with deep empathy—acknowledging both her emotional burden and the courage it took to share—but also engages her in a detailed, collaborative exploration of her experiences. By inviting her to pinpoint specific patterns and triggers behind her fears, Response 1 promotes self-exploration and empowerment, making it particularly effective for someone in the early stages of considering change. In contrast, the response of GPT-4o is general and less detailed, which can make the client feel less deeply understood. We provide more cases in Appendix 9.

---

**Client Speech**

I've been having emotional issues for a few years. Nonetheless, these have been somewhat manageable. However, I became increasingly paranoid this winter. I thought that people were trying to poison me, I feared that family members were going to kill me. I was aware that these thoughts were illogical, but I couldn't shake the fear that they caused me. I would get panic attacks thinking that I drank from a poisoned water fountain. I thought that someone had put small rips in my ice cream sandwich packaging because there was only one that didn't have tiny rips in it. I refused to eat this one because I assumed some reverse psychology was being pulled and the ones without the rips were fine. I thought that this was mostly caused by anxiety but there are other symptoms that lead me to believe it is something more.

---

**PsychoCounsel-Llama3-8B** (chosen)

I can sense the immense distress and fear that you've been experiencing, and I'm here to create a safe space for you to explore these feelings. It takes a lot of courage to acknowledge and share these thoughts with me. I want to acknowledge that it sounds like you've been carrying a heavy burden, and it's understandable that you felt the need to try to make sense of these experiences, even if they seemed illogical at times.
⇒ (detailed acknowledgment of the client's feelings tends to create a stronger emotional connection, making the client feel more profoundly understood)
I'd like to gently explore this further. You mentioned that you recognized these thoughts as anxiety-driven, but there are other symptoms that make you question whether it's more than that. Can you tell me more about what specifically makes you think it's not just anxiety? What are those other symptoms that have led you to consider this? Let's work together to tease out what might be contributing to these feelings, and see if we can identify any patterns or triggers that might be at play. We can also explore ways to help you manage these fears and develop more grounding techniques to reduce your anxiety. How does that sound?
⇒ (engage the client in a detailed, collaborative exploration of her experiences)

---

**GPT-4o** (rejected)

Thank you for sharing all of this with me. It sounds like you've been going through a particularly difficult time with these intense feelings and thoughts. It must be very distressing to experience such fears and the resulting panic attacks. I want to make sure we explore this in a way that feels safe for you.
⇒ (more general in validating the client's feelings)
You mentioned that while you've considered it might be anxiety, there are additional symptoms that make you think there might be more to it. Could you tell me more about these other symptoms? Understanding the full picture can help us figure out what might be happening and how best to support you.
⇒ (lacks the specific, collaborative invitation to explore the triggers)
Also, it's important to acknowledge the awareness you've shown about the thoughts being illogical at times, yet still powerful. This awareness is a valuable strength as we work through these experiences together. Let's take it one step at a time and see if we can find patterns or triggers for these thoughts and discuss some strategies that might help manage them.
⇒ (general and less detailed)

---

Table 4: Case Study

## 5 CONCLUSION AND FUTURE WORK

In this work, we introduce a set of professional and comprehensive principles for evaluating therapists' responses to client speeches in psycho-counseling, along with PsychoCounsel-Preference, a preference dataset containing 36k high-quality preference comparison pairs. Our experiments show that with PsychoCounsel-Preference, preference learning could effectively unlock LLMs' professional psycho-counseling skills. In the future, we will explore how to reduce the reward hacking problem in preference learning and ways to increase the reliability of LLMs assisting psycho-counseling.

## 6 ETHICS STATEMENT

This project has been classified as exempt by the Institutional Review Board (IRB). All hired experts were at least 18 years old and hold either a master's or doctoral degree in a mental health-related field, such as psychology or counseling psychology. Each expert received a fixed payment of $1,500 for all

annotations, corresponding to an approximate hourly rate of $60. The goal of this work is to leverage synthetic data and preference learning algorithms to equip LLMs with the skills needed to generate responses to client speeches in psycho-counseling. However, these responses should not be directly exposed to clients without review by real therapists. Instead, they serve as assistive suggestions to help therapists draft responses, improving the efficiency of psycho-counseling:

**Enhancing Human Expertise, Not Replacing It**. AI tools should function as supportive mechanisms that enhance the capabilities of mental health professionals rather than replacing them. These tools can provide therapists with valuable data-driven insights into client speech, suggest psycho-counseling responses, and assist in structuring interventions. However, their role remains assistive, ensuring human expertise remains central to patient care.

**Training and Ethical Integration**. The effective use of AI in psycho-counseling requires mental health professionals to receive specialized training. This ensures they can integrate AI-generated insights into their practice ethically and effectively, maintaining both professional oversight and adherence to best practices.

**Safeguards Against Unsupervised AI Interaction**. To uphold safety and ethical integrity, AI-generated insights and psycho-counseling suggestions should always be reviewed by a licensed professional before reaching a patient. Deployment models must include strict access controls, intervention thresholds, and supervision mechanisms to prevent autonomous operation without human oversight.

**Transparency and Accountability**

- Open Communication: AI deployment in mental health should involve clear and open communication with all stakeholders, including therapists, patients, and regulatory bodies. This fosters trust and ensures transparency in the development and use of AI tools.
- Explainability and Justification: AI-generated recommendations should be interpretable, providing clear reasoning behind decisions. This is particularly crucial for psycho-counseling suggestions and mental health assessments, where explainability is essential to professional trust and responsible use.
- User Awareness: Patients and therapists interacting with AI must be fully informed about the system's role, capabilities, and limitations to prevent over-reliance and misapplication.

**Safety and Privacy Standards**

- Error Mitigation: AI models should be rigorously tested to minimize the risk of errors in medical advice or psychological recommendations. Misdiagnoses or inappropriate interventions could have significant negative consequences.
- Preventing Misinformation and Hallucinations: AI systems must prioritize accuracy by reducing misinformation and hallucinations, ensuring responses are evidence-based and context-appropriate.
- Data Privacy and Confidentiality: AI tools in mental health care must adhere to strict data privacy regulations, ensuring that patient interactions remain secure and confidential. Compliance with legal and ethical data-handling standards is critical to protecting users from breaches or misuse.

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

## 7 PROMPTS

---

**Rating Prompt**

You are provided with a client speech and four responses from different psychotherapists. Rate the responses based on how they align with the given principle.

Client Speech: {client_speech}
Response 1: {response1}
Response 2: {response2}
Response 3: {response3}
Response 4: {response4}

Provide a JSON object as output that includes the following keys:
- response_1_rating: An integer score from 1 to 5 for response 1
- rationale_1: A string explaining the reasoning behind the given score for response 1
- response_2_rating: An integer score from 1 to 5 for response 2
- rationale_2: A string explaining the reasoning behind the given score for response 2
- response_3_rating: An integer score from 1 to 5 for response 3
- rationale_3: A string explaining the reasoning behind the given score for response 3
- response_4_rating: An integer score from 1 to 5 for response 4
- rationale_4: A string explaining the reasoning behind the given score for response 4

---

**Responding Prompt**

You are now a professional psychotherapist conducting a session with a client. Answer the given client speech.
Client Speech: {client_speech}

---

**LLM-as-Ranker Prompt**

Determine which of the two given responses from different psychotherapists to a client's speech is better:
Client Speech: {client_speech}
Response 1: {response_1}
Response 2: {response_2}

---

## 8 DATASET INFORMATION

**PsychoCounsel Principles**

**Empathy and Emotional Understanding**: The response should convey genuine empathy, acknowledging and validating the client's feelings and experiences.

- Emotional Reflection: Reflecting the client's emotions back to them.
- Validation: Affirming the client's feelings as legitimate and understandable.
- Non-Judgmental Tone: Maintaining a compassionate and accepting approach.

**Personalization and Relevance**: The response should be tailored to the client's unique situation, ensuring that the content is directly relevant to their concerns.

- Specific References: Mentioning details specific to the client's statements.
- Avoidance of Generic Responses: Steering clear of overly general or canned replies.
- Cultural and Individual Sensitivity: Respecting the client's background and personal context.

**Facilitation of Self-Exploration**: The response should encourage the client to reflect on their thoughts and feelings, promoting self-awareness and insight.

- Open-Ended Questions: Asking questions that invite elaboration.
- Reflective Statements: Paraphrasing the client's words to deepen understanding.
- Exploration of Thoughts and Feelings: Guiding the client to consider underlying emotions and beliefs.

**Clarity and Conciseness**: The response should be clear, well-organized, and free of unnecessary jargon, making it easy for the client to understand and engage with.

- Plain Language: Using words that are easily understood.
- Logical Flow: Presenting ideas in a coherent sequence.
- Brevity: Keeping the response concise while covering essential points.

**Promotion of Autonomy and Confidence**: The response should support the client's sense of control over their decisions and encourage confidence in their ability to make positive changes.

- Affirmation of Strengths: Highlighting the client's abilities and past successes.
- Encouraging Initiative: Motivating the client to take proactive steps.

**Avoidance of Harmful Language or Content**: The response should avoid any language or content that could potentially harm, distress, or trigger the client, ensuring the interaction is safe and supportive.

**Sensitivity to the Stage of Change**: The response should recognize the client's current stage in the process of change and address their needs accordingly. If the client is in an early stage—uncertain or ambivalent about making a change—the response should help them explore their thoughts and motivations. If the client is in a later stage and has already made changes, the response should focus on reinforcing progress, preventing setbacks, and sustaining positive

---

[9]https://mistral.ai/news/ministraux/
[10]https://mistral.ai/news/mistral-nemo/
[11]https://openai.com/index/openai-o1-system-card/
[12]https://docs.cohere.com/docs/responsible-use

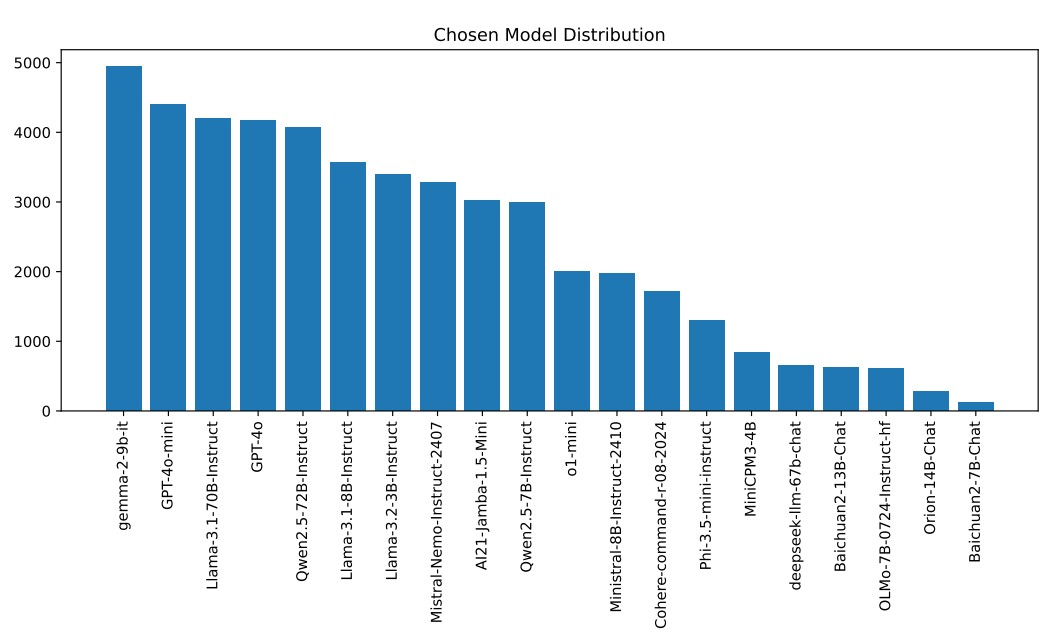

Figure 5: Chosen Model Distribution

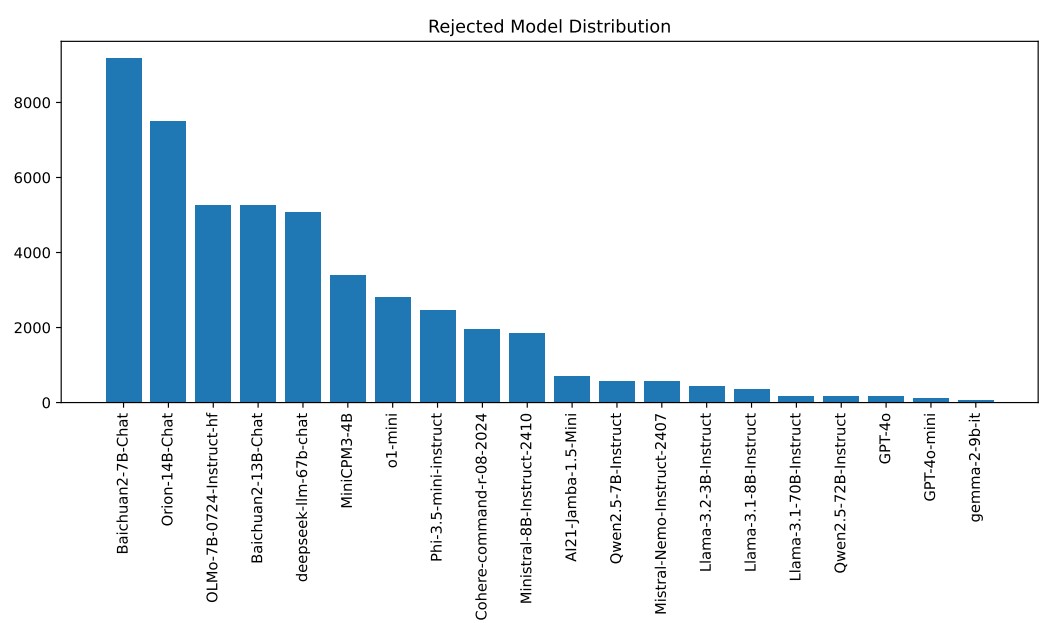

Figure 6: Rejected Model Distribution

Table 5: Topic Distribution

| Coarse Category | Fine Category | Count |
|---|---|---|
| **1. Core Mental Health Issues** | | |
| | Anxiety | 3714 |
| | Depression | 2859 |
| | Stress | 1439 |
| | Trauma | 526 |
| | Substance-abuse | 387 |
| | Addiction | 129 |
| **2. Emotional Well-being and Coping Strategies** | | |
| | Self-esteem | 1377 |
| | Grief-and-loss | 1023 |
| | Caregiving | 1541 |
| | Behavioral-change | 740 |
| | Anger-management | 448 |
| | Self-care | 311 |
| | Sleep-improvement | 277 |
| **3. Relationships and Interpersonal Dynamics** | | |
| | Relationships | 1690 |
| | Family-conflict | 2358 |
| | Friendship-conflict | 292 |
| | Marriage | 373 |
| | Intimacy | 403 |
| | Social-relationships | 410 |
| | Workplace-relationships | 383 |
| | Relationship-dissolution | 574 |
| **4. Life Transitions and Challenges** | | |
| | Career | 441 |
| | Aging | 140 |
| | New-environment | 235 |
| | Military-issues | 118 |
| **5. Social Issues** | | |
| | LGBTQ | 335 |
| | Culture | 113 |
| | Human-sexuality | 151 |
| | Bullying | 68 |
| **6. Youth and Development** | | |
| | Children-adolescents | 123 |
| | School-life | 322 |
| | Parenting | 730 |
| **7. Crisis and Safety Concerns** | | |
| | Domestic-violence | 144 |
| | Self-harm | 231 |
| | Eating-disorders | 154 |
| **8. Special Topics** | | |
| | Counseling-fundamentals | 638 |
| | Diagnosis | 531 |
| | Communication | 205 |
| | Professional-ethics | 128 |
| | Legal-regulatory | 94 |
| | Spirituality | 192 |
| | Others | 136 |

| Category | Models |
|---|---|
| **3-4B models** | Llama-3.2-3B-Instruct (Llama Team, 2024) |
| | Phi-3.5-mini-instruct (Abdin et al., 2024) |
| | MiniCPM3-4B (Hu et al., 2024) |
| **7-9B models** | Ministral-8B-Instruct-2410[9] |
| | Llama-3.1-8B-Instruct (Llama Team, 2024) |
| | gemma-2-9b-it (Gemma Team, 2024) |
| | Qwen2.5-7B-Instruct (Qwen et al., 2024) |
| | OLMo-7B-0724-Instruct (Muennighoff et al., 2024) |
| | Baichuan2-7B-Chat (Yang et al., 2023) |
| **12-14B models** | Baichuan2-13B-Chat (Yang et al., 2023) |
| | Orion-14B-Chat (Chen et al., 2024) |
| | Mistral-Nemo-Instruct-2407[10] |
| | AI21-Jamba-1.5-Mini (Jamba Team et al., 2024) |
| **65-75B models** | Llama-3.1-70B-Instruct (Llama Team, 2024) |
| | Qwen2.5-72B-Instruct (Qwen et al., 2024) |
| | deepseek-llm-67b-chat (DeepSeek-AI et al., 2024) |
| **Commercial models** | GPT-4o (OpenAI, 2024) |
| | GPT-4o-mini (OpenAI, 2024) |
| | o1-mini[11] |
| | Cohere-command-r-08-2024[12] |

Table 6: Overview of selected models in the pool.

## 9 CASE STUDY

In the example in Table 7, PsychoCounsel-Llama3-8B demonstrates a deeper level of empathy and personalization. It acknowledges the client's unique emotional experiences and suggests tailored strategies, such as the "3-4-5" exercise, to directly address their concerns. Additionally, it facilitates self-exploration by asking thoughtful questions about the client's fears, values, and understanding of self-trust, ultimately promoting autonomy and confidence in decision-making. In contrast, GPT-4o's response feels more clinical and general.

| Client Speech |
| --- |
| Anxiety often strikes when I'm faced with making decisions. The fear of making the wrong choice or disappointing others paralyzes me, leaving me stuck in indecision. I want to learn how to trust myself and make confident choices. |

| **PsychoCounsel-Llama3-8B** (chosen) |
| --- |
| It takes a lot of courage to share something so vulnerable with me, and I appreciate your trust in our conversation. I can sense the weight of that anxiety you're carrying, and we're going to explore ways to shift that together. |
| First, let's acknowledge that the fear of making the wrong choice or disappointing others is a common and normal feeling. It doesn't mean there's anything inherently wrong with you; it just means you care deeply about others and yourself. That's a beautiful quality to have. |
| Can you tell me more about what you think would happen if you made a "wrong" choice? What are some of the worst-case scenarios that come to mind? Sometimes, exploring these fears can help us gain clarity and perspective. We can work together to reframe those thoughts in a more realistic and compassionate light. |
| Let's also explore the idea of self-trust. What does it mean to you to trust yourself? What does self-trust look like in action? We can work on building that trust by identifying what you value and what aligns with your goals. We can practice recognizing when your thoughts are trying to control or second-guess you, and we can develop strategies to gently acknowledge those thoughts without letting them overwhelm you. |
| I'd like to introduce an exercise called "3-4-5." When you're faced with a difficult decision, I want you to take a deep breath and pause for 3 seconds, acknowledge the thought or feeling that arises, and then ask yourself: "Is this thought based on facts or perceived facts?" Sometimes, when we're anxious, our thoughts can get tangled up in assumptions or past experiences. Then, take 4 seconds to ask yourself what you truly value in this situation, what's most important to you. And finally, take 5 seconds to imagine yourself moving forward, taking action, and making a choice that aligns with your values. This exercise can help you tap into your inner wisdom and develop a sense of self-trust. |
| Lastly, let's work on reframing the idea of "wrong" choices. What if, instead of focusing on the fear of making the wrong choice, we reframe it as a learning opportunity? What if you saw every choice as a chance to learn, grow, and develop? How might that change your approach to decision-making? |
| Remember, our goal isn't to eliminate anxiety completely, but to develop a more compassionate relationship with it. We'll work together to cultivate self-awareness, self-acceptance, and trust in your decision-making abilities. |
| How do these ideas resonate with you? What feels most helpful or interesting to explore further? |

| **GPT-4o** (rejected) |
| --- |
| Thank you for sharing that with me. It's understandable that making decisions can be overwhelming, especially when there's fear of making the wrong choice or disappointing others. Let's explore this together. Firstly, it's important to acknowledge that decision-making can be challenging and it's completely normal to feel anxious about it. You're not alone in experiencing this. |
| One way we might start working on this is by exploring some of the underlying thoughts or beliefs that contribute to your anxiety about decision-making. For instance, you mentioned a fear of disappointing others. Can you tell me more about where you think that fear comes from? |
| We could also look at ways to build your confidence in your decision-making skills. Sometimes breaking down decisions into smaller, more manageable parts can make the process feel less daunting. Have you ever tried writing down the pros and cons of each option or talking it through with someone you trust? |
| Another technique is to practice self-compassion. Remind yourself that no one makes the right choice all the time and that it's okay to make mistakes. Every decision is a learning opportunity. |
| Finally, something that can be quite helpful is examining past decisions you've made. Often, we are harsher with ourselves than necessary and we overlook our strengths. Can you recall a situation in the past where you made a decision and things turned out well, or you learned something valuable from the outcome? |
| Let's take some time to explore these aspects and see what might resonate most with you. How does that sound? |

Table 7: Case Study - 2

