# OpenReview forum: "Preference Learning Unlocks LLMs' Psycho-Counseling Skills"
_ICLR.cc/2026/Conference — ICLR 2026 Conference Withdrawn Submission_

### Official Review · Reviewer_jBeh · 2025-10-30

**Soundness:** 2
**Presentation:** 3
**Contribution:** 2
**Rating:** 2
**Confidence:** 4

**Summary:**

This paper addresses the gap between the demand for mental health support and the availability of trained professionals by improving the psycho-counseling skills of LLMs. Recognizing that LLMs struggle due to a lack of high-quality, private counseling data , the authors first collaborated with experts to define a comprehensive set of "PsychoCounsel Principles" to evaluate responses based on seven dimensions, including empathy, relevance, safety, facilitation of self-exploration, and promotion of autonomy . Using these principles, they constructed PsychoCounsel-Preference, a new dataset with 36k high-quality preference pairs by scoring and comparing responses from 20 different LLMs to over 26k unique client speeches. Experiments involving reward modeling and preference learning on this dataset resulted in their best model, PsychoCounsel-Llama3-8B, which achieved an 87% win rate against GPT-40 , a preference that was also strongly supported by human expert evaluations. The authors will release their new dataset, the trained PsychoCounsel-Llama3-8B model, and the reward model to foster future research in this domain.

**Strengths:**

The main strength of this paper lies in its innovative methodology, which attempts to solve the challenge of lacking high-quality training data in the psycho-counseling field due to privacy constraints. The authors claim to have distilled professional counseling skills into a set of seven-dimensional evaluation principles through collaboration with experts in social work and psychiatry, and used this as a foundation to construct a large-scale preference dataset, PsychoCounsel-Preference. The paper reports strong experimental results, showing its trained PsychoCounsel-Llama3-8B model achieved an 87% win rate against GPT-40, a preference that was also supported by subsequent human expert evaluations. Finally, the authors promise to publicly release the dataset, reward model, and policy model, which is significant for promoting future research in this field.

**Weaknesses:**

This paper, while presenting a novel approach, suffers from several significant methodological weaknesses that call into question the validity and generalizability of its conclusions.

1. Reduction of Professionalism to a Fixed Rubric: The paper's core premise is that the complex, multi-faceted skill of "professional psycho-counseling" can be effectively reduced to seven quantitative dimensions. This is a significant reductionist leap. It is highly questionable whether these seven principles, scored in isolation, can adequately capture the tacit knowledge, strategic timing, and deep contextual understanding that define genuine therapeutic expertise.

2. Single-Turn Evaluation of a Dialogic Process: The data generation pipeline evaluates responses on a single-turn, context-agnostic basis . This fundamentally misunderstands the nature of psychotherapy, which is an inherently dialogic process evolving over time. A "good" response is not universally good; it is good at a specific moment. For instance, a "Cognitive Challenge" by a therapist may be crucial for a client's progress. Such a response would likely score low on "Empathy"  but be highly professional and effective. The paper's methodology, which averages scores across principles, would almost certainly filter out these critical but locally non-empathetic interventions, biasing the entire PsychoCounsel-Preference dataset towards "safe" and universally agreeable responses.

3. Incoherent Evaluation Loop: A major unresolved paradox exists in the results. The preference data was created using GPT-4o as the primary "evaluator" to score and pair responses. The final aligned model, PsychoCounsel-Llama3-8B, was then trained on this GPT-4o-generated data. It is therefore highly counter-intuitive and methodologically questionable that this model could achieve an 87.0% win rate against GPT-4o.

**Questions:**

How was GPT-4o beaten by a model trained on its own preference data? Does this counter-intuitive result reveal an inherent instability in the LLM-as-a-judge method?

---

### Official Review · Reviewer_FcEx · 2025-10-30

**Soundness:** 2
**Presentation:** 3
**Contribution:** 2
**Rating:** 2
**Confidence:** 4

**Summary:**

Using these principles, the authors constructed the PsychoCounsel-Preference dataset. They first collected around 26,000 client statements across 8 categories and 42 subtopics. Then, they generated responses using 20 large language models. Each response was scored using GPT-4o, based on the seven principles. Pairs with clear score differences were selected, resulting in about 36,000 preference samples. Two counselors validated 200 samples, showing 88.5% agreement with the dataset labels.
The authors used this dataset to train both a reward model and a policy model. The reward model, based on LLaMA3 (3B and 8B), reached 97–98% accuracy on the test set. The policy model was trained using offline and online Direct Preference Optimization (DPO). The final PsychoCounsel-LLaMA3-8B model achieved a win rate of 87% (unrestricted length) and 77% (restricted length) compared to GPT-4o. Ablation studies showed that online DPO improved stability and helped smaller models approach the performance of larger ones.

**Strengths:**

This paper clearly combines ideas from psychological counseling with LLM-based preference learning. It fills a real gap in the field. Moreover, the paper explains the seven evaluation principles, grounding them in solid psychological theory and applying transparent scoring rules.
The dataset is large and built using a solid method. The experiments cover a wide range. A notable finding is that small models trained online can perform nearly as well as larger ones, which is useful for real-world use.
The paper is well-organized. The figures and tables are clear and helpful. Also, sharing the code and data openly will support future research and help grow interest in this area.

**Weaknesses:**

The main issue is with the evaluation setup. GPT-4o is used both to create training labels and to test the model. This creates a feedback loop. The model might just learn to copy GPT-4o’s style instead of showing real counseling ability. So, the high 87% win rate probably reflects this stylistic match, not actual ability.
The human evaluation is also limited. It only uses 2 reviewers and 200 samples. Talos, there’s no mention of how much the reviewers agreed with each other, which makes the results less reliable. Also, it’s not clear how the “win rate” relates to real counseling quality or human judgment.
The model was not tested on other datasets. There’s no comparison with existing mental health tools or checks by outside experts. The title suggests ‘stronger counseling ability’, but the tests only look at single-turn responses. That’s a mismatch, since real counseling often depends on longer conversations.

**Questions:**

The main results rely entirely on GPT-4o’s own judgments. It’s unclear whether the model would perform as well if evaluated by other systems, like Claude, or by human experts. Adding evaluations from other models or independent human raters would help confirm that the performance isn’t just reflecting GPT-4o’s preferences.
Right now, the human evaluation includes only two counselors and 200 sample pairs. That seems too small to fully support the paper’s main claims. A larger study and a measure like Cohen’s κ would strengthen the results. It would also help to explain how each counseling principle contributes to the final score.
Finally, the model hasn’t been tested on high-risk cases like suicide, self-harm, or medical issues. These are critical in mental health. Adding safety tests or at least discussing the risks and limits would make the paper stronger.

---

### Official Review · Reviewer_UoYY · 2025-11-01

**Soundness:** 3
**Presentation:** 3
**Contribution:** 2
**Rating:** 4
**Confidence:** 3

**Summary:**

The paper constructs a dataset, PsychoCounsel-Preference, to train LLMs for psycho-counseling via preference learning. Results indicates that the trained models win over GPT-4o 87% of the time.

**Strengths:**

1. Trained policy models show a nice win-rate over GPT-4o, both by the reward model and by human annotators.
1. The dataset is constructed with principles, which potentially make the dataset more rigorous.

**Weaknesses:**

1. The level of agreement (88.5%) can be placed in a better context for readers to evaluate the significance of it. What is the standard or acceptable threshold in the therapy community? What is the average inter-human agreement (two random therapists is just too small to make 87% a trustable standard)? How to know if this number is good enough? How to know if it reaches a standard that makes the dataset useful?
1. The human annotation only provided a measure but not control of the quality of the dataset
1. Evaluation scale is a bit limited.

**Questions:**

1. "The two therapists agree on 174 out of 200 samples. Additionally, one expert’s annotations align with the preference labels in PsychoCounsel- Preference for 184 out of 200 samples, while the other aligns for 170 out of 200 samples" This piece of text sounds confusing to me. How do these numbers mean compared to each other? I am confused because 174 is not an average nor an intersection of the two, so I wonder what the number of sample "two therapists agree on" indicate here.
1. Any study on whether the principles proposed in the paper quantitatively improves the quality of the dataset?

---

### Official Review · Reviewer_TJnN · 2025-11-13

**Soundness:** 1
**Presentation:** 2
**Contribution:** 2
**Rating:** 4
**Confidence:** 2

**Summary:**

This paper presents PsychoCounsel-Preference, a benchmark dataset of preference comparison pairs for psycho-counseling tasks. The authors develop quality evaluation principles with domain experts (social workers and psychiatrists), construct preference pairs from existing counseling datasets, and validate them with professional psychotherapists. The paper evaluates existing reward/policy models and models trained on PsychoCounsel-Preference, reporting performance improvements. Ablation studies and qualitative analyses are included to demonstrate dataset utility.

**Strengths:**

This paper tackles a meaningful problem at the intersection of NLP and mental health, where high-quality evaluation resources are scarce. It includes comparisons across multiple models (reward models, policy models, and LLMs) with ablation studies and qualitative analysis. PsychoCounsel-Preference might provide a concrete resource that can advance research in AI-assisted counseling applications.

**Weaknesses:**

The PsychoCounsel principles lacks adequate grounding and validation. While developed by domain experts, the paper does not explain and answer the considerations such as 1) How these principles were systematically derived 2) How they connect to established counseling effectiveness frameworks in psychology/social work literature 3) Their relative importance or weighting in evaluating counseling quality 4) Whether they were empirically validated beyond face validity from two psychotherapists. Given that the entire dataset is constructed based on these principles, their theoretical and empirical foundation is critical. The paper would benefit by relating the principles to existing therapeutic frameworks (e.g., person-centered therapy, CBT principles) and demonstrating correlation with counseling outcome measures if available.

**Questions:**

Please refer to the Weaknesses section to clarify the concerns and questions. I hope to listen to the authors' response.

---

### Note · Authors · 2025-12-15

I have read and agree with the venue's withdrawal policy on behalf of myself and my co-authors.